# Exploiting HBIM for Historical Mud Architecture: The Huaca Arco Iris in Chan Chan (Peru)

Francesca Colosi [1,*], Eva Savina Malinverni [2], Francisco James Leon Trujillo [3], Roberto Pierdicca [2], Roberto Orazi [1] and Francesco Di Stefano [2]

[1] Consiglio Nazionale delle Ricerche (CNR), Istituto di Scienze del Patrimonio Culturale, Area della Ricerca Roma 1, Via Salaria km. 29,300, 00016 Rome, Italy; roberto.orazi@ispc.cnr.it

[2] Dipartimento di Ingegneria Edile, Civile e Architettura, Università Politecnica delle Marche, Via Brecce Bianche, 12, 60131 Ancona, Italy; e.s.malinverni@staff.univpm.it (E.S.M.); r.pierdicca@staff.univpm.it (R.P.); f.distefano@staff.univpm.it (F.D.S.)

[3] Carrera de Ingeniería Civil, Instituto de Investigación Científica, Universidad de Lima, Avenida Javier Prado Este 4600, Lima 15023, Peru; fleon@ulima.edu.pe

* Correspondence: francesca.colosi@cnr.it; Tel.: +39-069-067-2689

**Abstract:** The construction technique of raw earth, which has always been in use in most of the world, has left large monuments or architectural complexes to cultural heritage that need special attention due to the notable vulnerability of the material. A convenient way to deal this threat, besides physical intervention, is by using an information system, such as HBIM (Heritage Building Information Modeling), as a tool for damage assessment and conservation planning. This paper reports on its application in an archaeological setting, in particular, on the Huaca Arco Iris, a religious building of the old city of Chan Chan (Peru), the largest monumental complex in mud on the American continent. The study is part of the bilateral international project between the Consiglio Nazionale delle Ricerche (CNR) and the Consejo Nacional de Ciencia, Tecnología e Innovación Tecnológica (CONCYTEC) in the use of HBIM for the prediction of possible natural or anthropogenic damages to buildings in raw mud. Exploiting the data coming from the direct and indirect analyses, a dedicated ontology is built to guide the management of these data within the information system. The creation of an HBIM system for the archaeological domain, based on the trinomial data–information–knowledge, is presented and validated. Following this approach, a customizable HBIM has been created with the 3D model of the spatial entities of the Huaca. As a result, the semantic relationship of an external wall, taken as the benchmark test of our experiment, with the contained bas-relief and the conservation cover is tested.

**Keywords:** mud architecture; archaeological site; Peruvian heritage; photogrammetry; 3D model; ontology; HBIM; information management system

## 1. Introduction

### 1.1. Mud Architecture

Among the most ancient and widespread building materials in the world is undoubtedly the one made with clay and straw. This material can be kneaded with water and laid through formwork (*pisé*) or supported by wooden frames or reeds (*torchis*) or even through bricks of various sizes dried in the sun (*adobe*) [1–3]. Its versatility makes it the material with which, to date, the homes of about a quarter of the world's population are built and its workability is such that more than 180 sites of the UNESCO World Heritage List consist of architectural and/or archaeological artefacts in raw mud [4,5].

The attention towards architecture in *adobe* and its multiple artistic manifestations has been growing during the last fifty years, in particular, since the realization of the first international conference on the theme was held in Yazd (Iran) in 1972. A few years before, it had already prompted a discussion on the practice of architecture as design and

drawing as opposed to more or less spontaneous construction. Around those years, it was still preferable to talk about vernacular architecture or architecture without architects [6]. Although today earthen constructions are widely perceived as "mud huts" associated with an image of poverty and social and cultural marginalization, their significance and potential are known and recognized. Available in nature at an extremely low cost, earth requires a reduced energy cost for its processing and offers the opportunity to construct building with a remarkable living comfort thanks to its characteristics of breathability and thermal insulation [7,8].

In recent years, environmental considerations have become increasingly important in Europe (Dachverband Lehm, CRAterre, etc.) as well as internationally. Building materials that are environmentally friendly, energy-adjustable, sustainable and recyclable are experiencing a real renaissance.

However, increasingly threatened by natural and human impacts (e.g., floods and earthquakes, industrialization, urbanization, modern building technologies, disappearance of traditional conservation practices, etc.) mud structures deserve particular attention in terms of conservation and maintenance. About a fifth of the sites inscribed on the World Heritage List in Danger are earthen sites [9,10].

Since the 1990s, the conservation of mud architecture and related problems has been the subject of studies and interviews by the Getty Center, which in 1994 organized, in collaboration with ICCROMM, the first Pan-American course on the subject, precisely in *Museo de Sitio* of Chan Chan (Peru) [11].

Given the importance acquired by the theme over the years [12–15], the same UNESCO, in collaboration with CRATerre-EAG (International Centre for Earth Construction–School of Architecture of Grenoble and the most important reference at world level), has promoted the World Heritage Programme on Earthen Architecture (WHEAP). The program is addressed to the formulation of specific research projects aimed at improving conservation techniques and methods of management of monumental complexes on earth.

As UNESCO states, "Expected results include a better understanding of the problems facing earthen architecture, the development of policies favoring its conservation, the definition of practical guidelines and the organization of training and awareness activities, particularly in local communities through workshops, exhibitions, conferences and technical support publications to raise the recognition of earthen architecture, as well as the creation of an active global network for the exchange of information and experience" [16].

### 1.2. Research Purposes and Aims

Chan Chan, the capital of the Chimor empire (IX-XV cen. AD), is the largest pre-Columbian city built in mud. It is located about 600 km from Lima (Figure 1), near Trujillo, the third largest city in the country, and extends over fourteen square kilometers with articulated architectural structures surrounded by imposing walls, remnants of popular houses, and structures of worship in the shape of stepped pyramids named *huacas* (Figure 2) [17–21]. Due to the extraordinary example of its urban organization and the precious testimonies of the costal pre-Inca cultures, Chan Chan was inscribed in the UNESCO World Heritage Site in 1986.

As required by UNESCO for all World Heritage Sites, Chan Chan has a very complex and articulated management plan, the *Plan Maestro de conservación y manejo del Complejo Arqueologico Chan Chan (Plan Maestro)*, drawn up by the Instituto Nacional de Cultura (today, Ministerio de Cultura) and approved by the Peruvian government in the year 2000. In recent years, the Plan Maestro has been revised and updated thanks to the collaboration between Direccion Desconcentrada de Cultura–La Libertad and the Proyecto Especial Complejo Arqueologico Chah Chan (PECACH), a special unit which has been operating in Chan Chan with economic and managerial autonomy since 2006 [22].

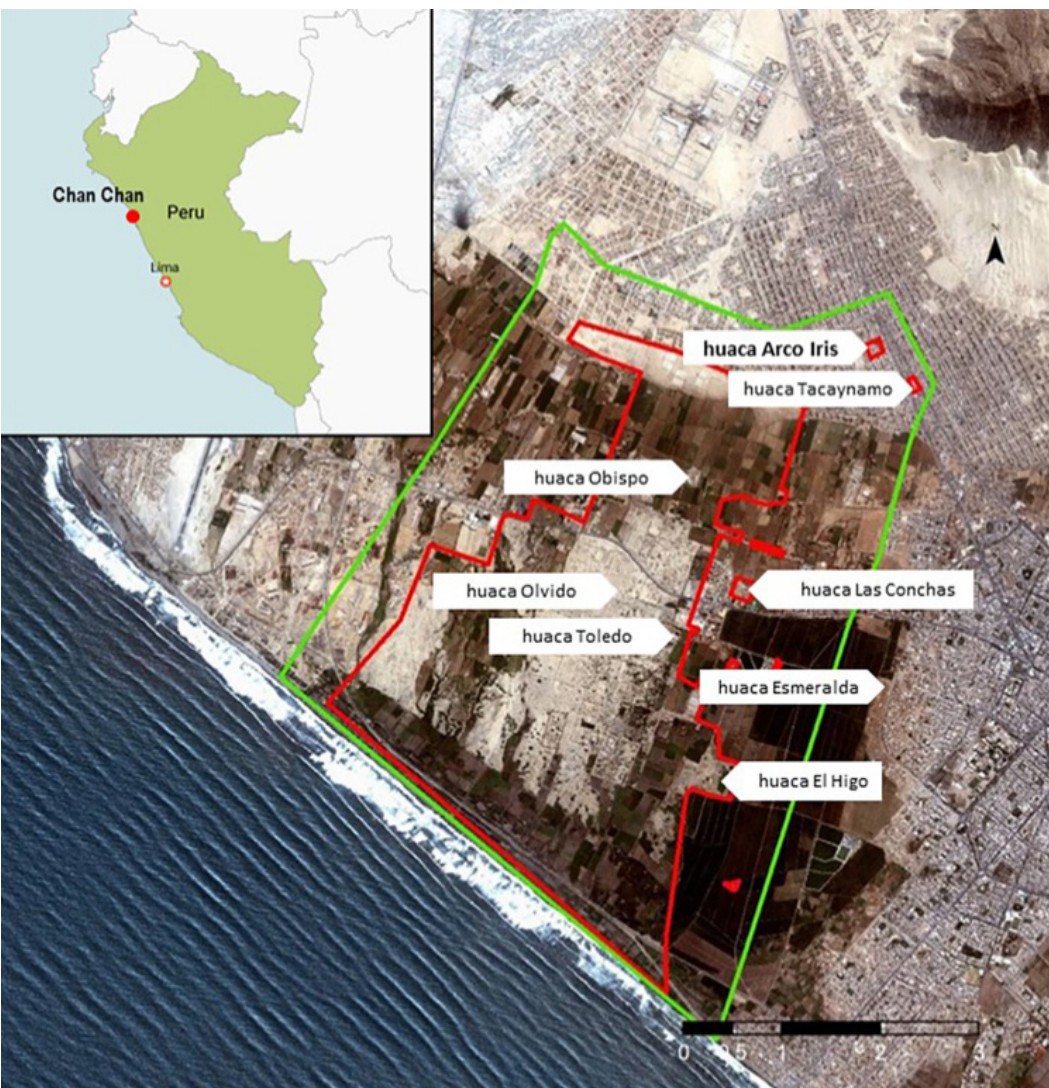

**Figure 1.** The archaeological area of Chan Chan (8°04′35″ S; 79°02′55″ W) with the indication of Huaca Arco Iris and the other *huacas*. The core zone is drawn in red, the buffer zone in green.

A chapter of the Plan Maestro 2021–2031 foresees the qualitative and quantitative identification of the materials used in the buildings of Chan Chan, establishing their physical–mechanical behavior and their influence, determining the optimal characteristics and percentages in the elaboration of mortars or mixtures. In fact, it is fundamental to know the materials and their percentages used in the elaboration of the different clay mortars (for adobes, plasters and reliefs) and in the different construction phases.

This study has been carried out by the CNR-Institute of Heritage Science, Università Politecnica delle Marche and Universidad de Lima as part of the joint research project "Plan for the management of the risk of damaging events in Chan Chan (Trujillo, Peru): multidisciplinary analysis and HBIM methodology", launched on the basis of the bilateral agreement CNR-CONCYTEC 2021–2022. The main objective of the project is to support the Plan Maestro 2021–2031 with the realization of a risk prevention plan on the monumental complex using the GIS 3D and HBIM methodologies in an integrated way. The experimentation is carried out on some architectural and urban structures of Chan Chan, on which a vast work of documentation with traditional methods has been completed, to formulate a project to identify the atmospheric, geological or anthropogenic risks and design any preventive means of protection or safeguard.

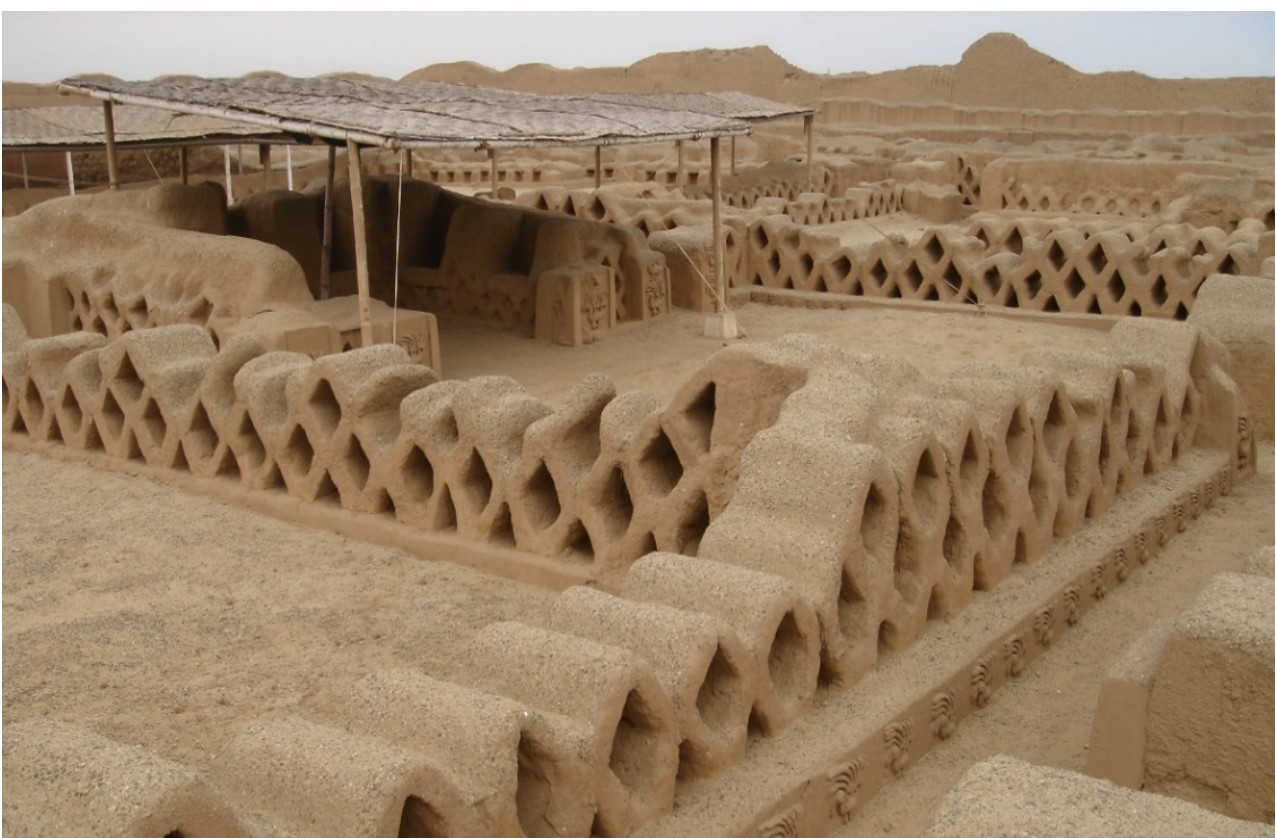

**Figure 2.** Chan Chan. A view of the buildings in *adobe* (Palacio Tschudi). Photo by R. Orazi.

In this context, Huaca Arco Iris represents an important case study for the Italian archaeological mission in Peru (MIPE), which in the past years had carried out a photogrammetric survey of the monument.

The research on Huaca Arco Iris aims not only to consolidate the use of BIM for the documentation, analysis and conservation of the built heritage, whether it is an architectural or archaeological structure, but in particular, seeks to test and enhance the use of BIM in the management of historical heritage built in raw earth. Given the importance and the diffusion of this specific construction technique, the use of BIM could be particularly useful to handle conservation projects in an agile and comprehensive way on a large part of the cultural heritage, especially in the areas of the American continent and the Indo-Asian regions.

### 1.3. BIM for the Archaeological Heritage

Geomatics techniques and methodologies are nowadays paramount, as they represent the starting point to study, manage and preserve archaeological monuments. The current wide availability of digital tools and media offers new opportunities for registering, analyzing and disseminating archaeological data and information. Although new tools have changed the way archaeological findings are managed in digital format, the sole 3D geometric reconstruction is not enough in the whole process of digitization. It is necessary to include relevant information to take advantage of 3D models for the physical and digital management and preservation of archaeological sites and monuments. The Archaeological Building Information Model (A-BIM) is considered a stable research field in geomatics and has proven to be a turning point in changing the methodological approach of the documentation and analysis phases [23,24]. The term A-BIM has recently been coined to refer to an information system for the management of archaeological buildings. It is distinct, in term only, from HBIM, which refers to historic buildings that are part of the built heritage

or to existing buildings. This information management platform has great potential, as it allows generation of 3D models and associating with them various information in the form of attributes, such as restoration works carried out over time, type of raw material or assumptions from documentary sources and much more. In this way, a BIM system for archaeology becomes a database containing information available to different actors, stakeholders, in relation to an artifact type. It then becomes an integral management, conservation and planning tool for assessing the structural condition of the building and the preservation of its components.

In [25] the authors undertook an archaeological survey of the Sun Temple of Niuserre at Abu Ghurab, Egypt. The authors of [26] did not cite the acronyms verbatim, but they did integrate their BIM work into what they call an "archaeological framework", using the ARCHAEOBIM terminology, and [27] modelled the Marzabotto Tuscan temple from drawings.

Information plays a key role in BIM workflows and can be of different types depending on research needs. In this sense, information concerning archaeological buildings could be different, for example, from land concessions to written archives, or from historical pictures or paintings to past restorations and human memories. A-BIM is a novel and pending domain that could avoid data fragmentation derived from traditional studies, establishing a watershed between this new court of documentation through BIM methodologies and traditional archaeological analysis. The latter is often characterized by paper-based works that should be linked with the metric and architectural context and with other digital studies [28–30].

However, the integration of information not directly designed to be included inside a BIM platform is still a critical problem as well as a major academic challenge, especially since it also depends on testing experimental and unconventional ways to integrate this kind of historical data on BIM platforms [31–33].

HBIM is defined as a particular type of BIM that handles the digitization and modeling of heritage/historic buildings [34,35]. HBIM has its specificities, e.g., it is generally created through a 3D survey by using massive in situ data acquisition techniques [36]. This implies processing large volumes of data that need to be interpreted and parameterized as BIM components following engineering and architectural logic and rules. This is complicated further in the archaeological field, and an A-BIM implementation requires a customized approach for the uniqueness of elements. Despite being challenging, the benefits of an A-BIM are also well-known [37], from visualization to monitoring and documentation, highlighting the potentiality of the usual centralized database architecture that BIM solutions follow. Our research contributes in this direction by applying the BIM platform for the analysis and management of historic earthen buildings. It is completely innovative for Latin America, though this method has been used in Europe for several years and is multiplatform. The 3D BIM model presents considerable complexity due to the characteristics of this type of spontaneous and often homemade architecture. Materials, techniques and methods of construction ensure that the clay structures never follow defined canons in typology and size. The easy perishability of the mud brick causes the deterioration of the monuments, depriving them of defined block references and transforming them into rounded and washed-out structures.

The proposed approach includes semantics from an ontological point of view; in the following this data structure is described.

## 2. Materials and Methods

This section is devoted to summarizing the research activity conducted to implement an information management system through BIM on an example of ancient mud architecture. The case study of Huaca Arco Iris is introduced in the Section 2.1.

The methodological workflow is structured based on the trinomial *data–information–knowledge* (Figure 3).

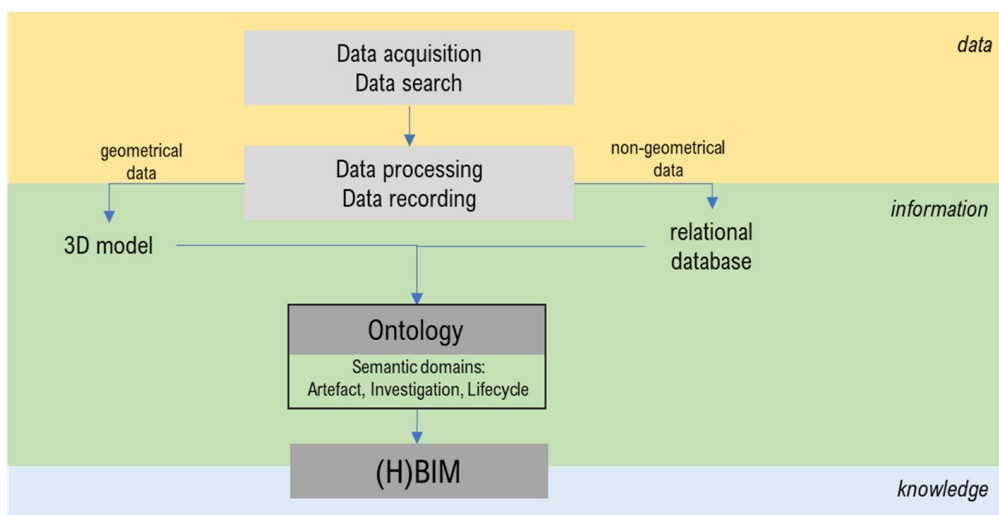

**Figure 3.** This schema depicts the trinomial data–information–knowledge representing the baseline of our workflow.

- *Data*. First, geometrical data are acquired with the use of geomatics techniques. In addition, historical research and condition state assessment are carried out. After the following step of data processing and recording, a 3D model and relational database are ready to be created.
- *Information*. An ontology schema is needed to manage the information in the form of 3D shapes of the architectural objects and the non-geometrical data linked to them. The ontology is structured in several domains: *artefact* related to the functional breakdown of the architectural elements, *investigation* based on direct and indirect analyses of the heritage site, *lifecycle* referring to the condition assessment of the architecture and actors involved.
- *Knowledge*. The definition of the dedicated ontology is useful to guide the implementation of a knowledge modeling through a BIM platform. Indeed, BIM is intended as software based on the parametric modelling and not characterized by an ontological approach. An HBIM application is realized here, where the features represent the 3D entities of the heritage and the relationships between them are defined according to the ontology schema.

### 2.1. Case Study: Huaca Arco Iris in Chan Chan

In the Quechuan languages of South America, a *huaca* is an object that represents something revered, typically a monument of some kind. The term *huaca* can refer to natural locations, such as immense rocks. Some *huacas* have been associated with veneration and ritual.

The *huacas* of Chan Chan (Trujillo, Peru) are located at a certain distance from each other and distributed evenly throughout the territory. To the north, the great Huaca Obispo; to the east, the Huaca las Conchas and the Huaca Esmeralda, both outside the protected area; further south, the Huaca Olvido and the Huaca Toledo, which in recent years has been completely unearthed and restored; and finally to the southeast, the Huaca El Higo.

The Huaca Arco Iris is located in La Esperanza District to the north of and outside the core zone, that is, the protected archaeological area of Chan Chan (Figure 4). Due to the urban development of the nearby Trujillo, it is surrounded by recent houses from which it is separated thanks to a wall that protects it from the urban development.

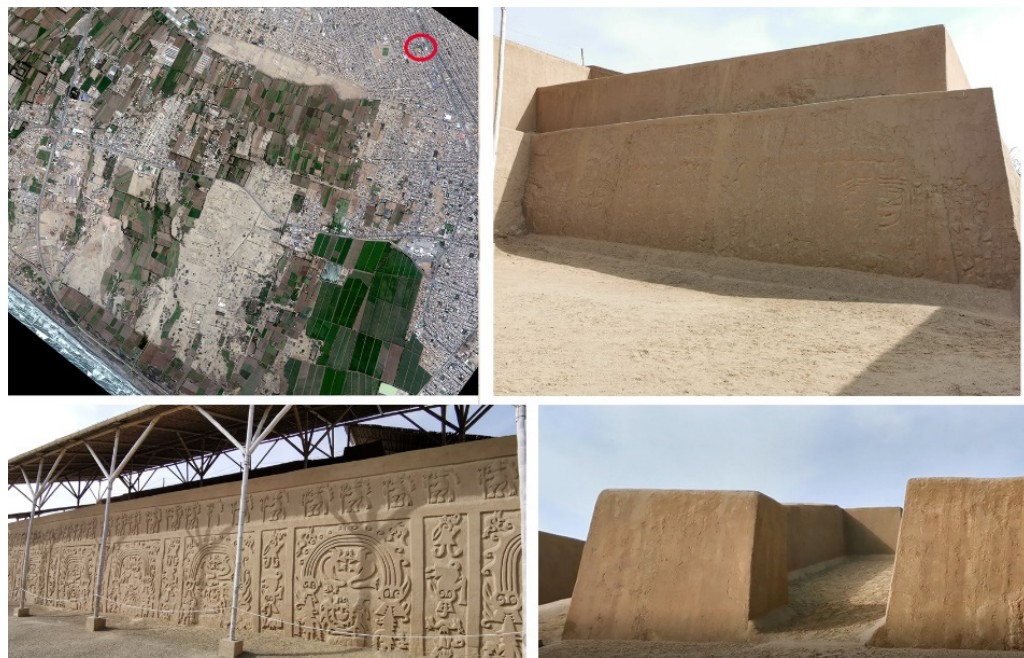

**Figure 4.** On the left: localization of the Huaca Arco Iris (**top**) and external south wall (**bottom**). On the right: view of the huaca (**top**) and of the ramp (**bottom**).

Inside the modern enclosed area, the Huaca Arco Iris has a wall enclosure covered with bas-reliefs that surrounds the sacred building itself. The latter, to which probably only the priest could access, consists of a truncated pyramid with two different levels. The first compact platform, with an area of 777 m$^2$ and a height of 4.50 m, is the most important part of the building. This leads to a smaller platform, built on the previous one, of 366 m$^2$ and 3 m high. A ramp goes up to the first platform and to a small hall from which it is possible to access, again via a ramp, the upper terrace. Along three sides of the first platform are aligned 14 cubicles or warehouses 3 by 2 m, whose external walls are covered by large bas-reliefs that have been interpreted as a representation of the rainbow (*Arco Iris*) overarching two fantastic animals that emit flames from their mouths (dragon-like creatures) from which the second name (Huaca del Dragón) of the sanctuary derives [18,19].

According to scholars of pre-Hispanic cosmogony, the rainbow was seen as a catastrophic event, a phenomenon of which the population was terrified especially because they feared that it could enter their bodies and bring misfortune or disease [38]. Hence, probably, the association with hideous animals emitting flames or dragons. This iconography is similar to that of Huaca El Higo and Huaca Chotuna in Lambayeque, which, according to Donnan [39], commemorate a strong El Niño event.

Chronologically, Huaca Arco Iris corresponds to the first buildings of Chan Chan (Early Phase), at the end of the Middle Horizon of the relative chronology of the central Andes. Its function seems to have been that of a royal funerary platform where sacral ceremonies periodically took place. The peripheral enclosures without access were apparently used as cubicles for the repository of funerary goods and perhaps the companions of the deceased.

In its current form, the monument is the result of a full restoration work that was performed in 1963 by the Patronato de Arqueología of Trujillo. Most of the bas-reliefs were restored with integrations or with the creation of parts without decoration where it was preferred not to intervene with the creation of new shapes.

The archaeological work, carried out by the Patronato de Arqueología, revealed the existence of color in the reliefs and walls of the huacas Arco Iris and Taycanamo (District of La Esperanza). It would be important to identify the pigments of the decorative system, establishing their physical–mechanical behavior and their compatibility with clay surfaces also, in order to create a colored image of the *huaca*.

The main dangers to the maintenance of the building stem from the catastrophic effects of the Niño that, in some years, have caused leaching of the walls and bas-reliefs and that are sometimes resisted by building a temporary rain protection structure.

Within the framework of the ENSO (El Niño Southern Oscillation) prevention program, undertaken between the end of 2014 and 2016, protection work was carried out by PECACH) in the most vulnerable sectors of the *huaca*. No chemical damage was detected. Perhaps the distance from the sea and the aquifer has prevented, for now, damages that were detected as water infiltrations from the ground on the vertical structures, or presence of efflorescence.

According to a first analysis of the construction elements, we can identify the two platforms, the vertical elements or load-bearing walls, the floors, the inclined elements or ramps, and finally, the bas-reliefs for the decorative aspects. Although for the construction technique of the inclined elements we cannot have precise indications, the vertical elements and the platforms were built using *adobe* of different sizes made with mud mortar and finally covered with mud plaster or bas-reliefs carved on the spot.

### 2.2. Survey Data Acquisition and Processing

To obtain the geometric data for the 3D modeling of Huaca Arco Iris, a metric survey was carried out using the spherical photogrammetry technique. The survey was performed in 2016 and is described thoroughly in a previous publication [40,41]. This made it possible, during the restitution phase, to model not only the volumetry of the historical monument (Figure 5), but also details such as the beautiful bas-reliefs preserved today (Figure 6). From this previous work, a 3D wireframe model was produced where the blocks making up the volumetric structure of the huaca are represented by polylines and surfaces. Based on this starting data, it was decided to adjust the volumetry of the huaca by defining solid blocks to be semantically broken down based on the defined ontology and for subsequent modeling in the BIM software.

During the last MIPE mission completed in 2018, an inspection was carried out at the Huaca Arco Iris site. A protective adobe brick cover was found in place of a bas-relief on the south-eastern external wall of the *temenos*. In order to document this, a further photogrammetric survey was performed to reconstruct a 3D model of this conservative cover. A digital camera (Sony Alpha77) was used to take 43 photos, ensuring overlapping images, at a distance of 3.50 m from the wall. These images were processed through Metashape software, which generates as output a textured mesh of the wall with the protective cover (Figure 7).

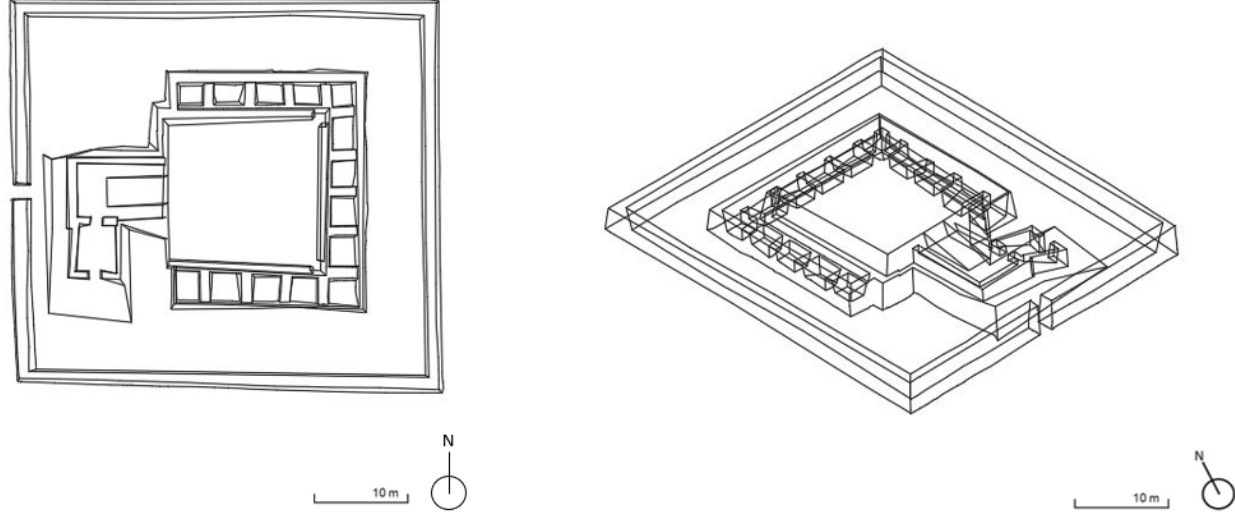

**Figure 5.** Wireframe model (plan and axonometric view) of the *huaca* obtained after restitution phase with spherical photogrammetry [40].

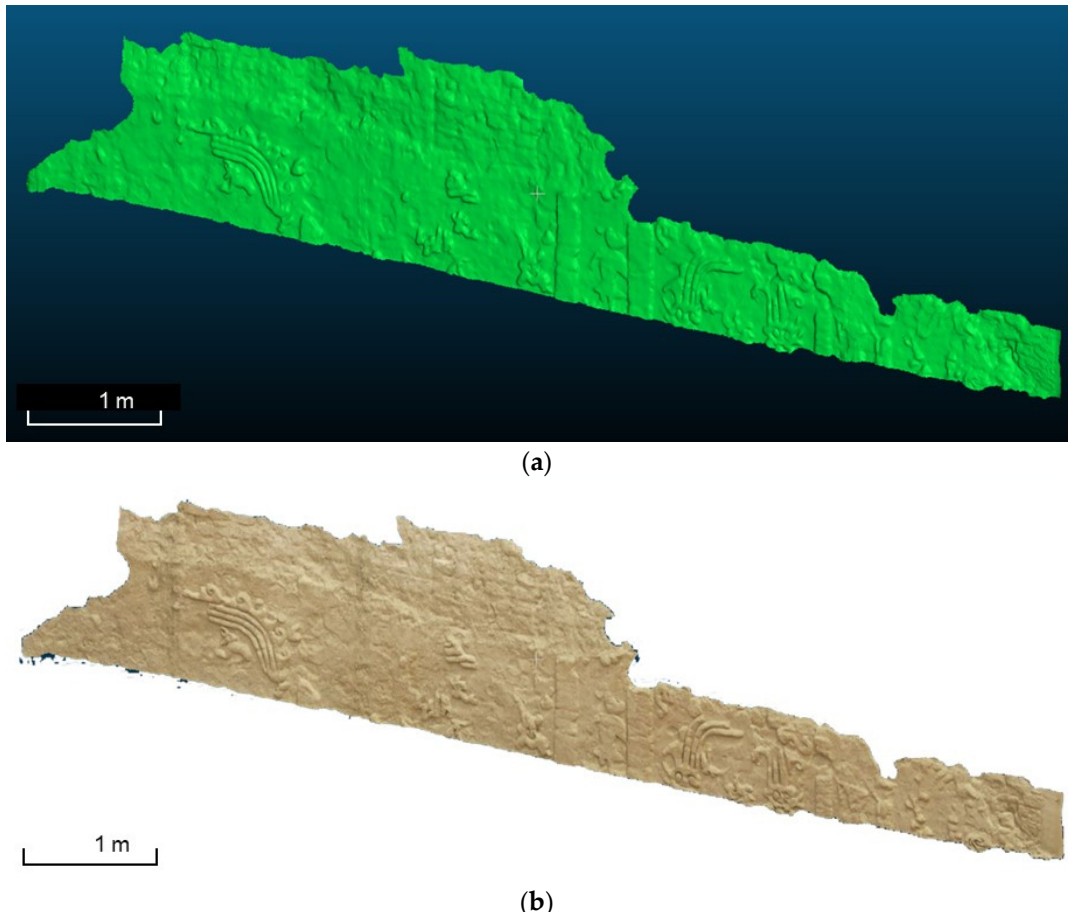

(**a**)

(**b**)

**Figure 6.** Elaboration of a bas-relief model: (**a**) mesh and (**b**) textured mesh.

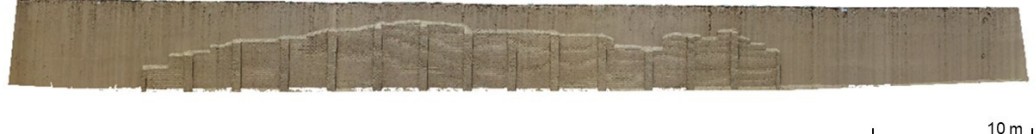

**Figure 7.** Textured mesh of the protective adobe brick cover (south-west external wall).

### 2.3. Dedicated Semantic Ontology

An ontology is defined as an interpretative process to collect information about a particular object in orderly and logically linked forms [42]. Today, ontologies have found a wide range of applications. The analysis of the state of the art indicates previous research aimed at establishing an ontology-based framework for modeling architectural knowledge.

The approach to artefacts (survey, modeling and data enrichment) should be based on knowledge as a guiding principle; in this sense it is possible to talk about 'knowledge modeling', as [43] have recently affirmed. In this perspective, this work aims to develop a strategy for architectural knowledge modeling in order to actively support the documentation and conservation process for a particular example of mud architecture. A two-fold approach to synchronize two different ways of representing reality is needed, which do not automatically interact: one ontology-based to represent the semantic non-geometrical information as data regarding building history and state of conservation, and one in the BIM environment that is absolutely suitable to represent the logic of the construction, above all its geometrical-constructive aspects.

A general ontology referring to a built heritage can be schematized on the basis of four main semantic domains:

- Artefact: spatial and construction entities composing the built heritage object.
- Investigation process: direct analysis of the built heritage object through metric survey, material analysis, diagnostic analysis and decay analysis; and the indirect analysis based on historical and archival search, bibliographic search, and also sources from local communities and evaluations of experiences by site visitors.
- Lifecycle: the historical background of the built heritage object in the form of transformation and changes in construction phases and the actual conservation state assessment;
- Actors: experts involved in the HBIM project of the built heritage object (for example, academics, engineers and architects, or archaeologists). Stakeholders are also considered, such as public administrations, from Peru's Ministry of Culture to the municipality of Trujillo. Other figures can be included, for example, future users of the archaeological area (visitors and residents). We seek to disseminate knowledge of the artefact to them through a web-sharing platform of the implemented HBIM project.

The ontological schema is commonly represented as a graph structure that refers to the logical composition of the conceptual map, also taking into account the relationships between the elements with reference to the relational database. These semantic domains thus make it possible to manage heterogeneous data from the same built heritage (Figure 8).

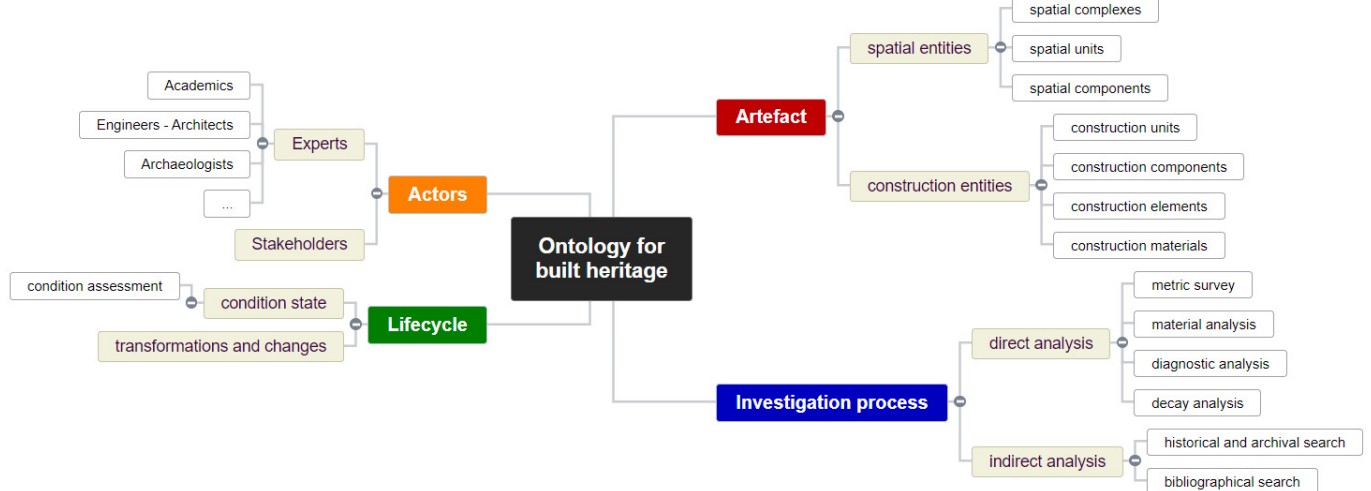

**Figure 8.** Ontology schema for built heritage object.

Once the survey and analysis of the state of conservation of the *huaca* were completed, a specific ontology to adequately represent the historical architecture was formalized. This ontology aimed at reproducing, as consistently as possible, the logical structure characterizing the documentation and conservation process in the HBIM environment. The idea was therefore to formalize the knowledge of the built heritage by reproposing, on a conceptual level, all propaedeutic steps for understanding, preserving and managing the heritage site in the medium-long term. Since the architectural and functional structure of the *huaca* are simple, the ontological schema referring to the artefact domain is structured as in Figure 8.

The diagram shown in Figure 9a illustrates the ontological aspects addressed in this project with reference to the implementation of the HBIM model of huaca. According to the artefact, the spatial elements and constructive components that are modeled in BIM environment are indicated. Figure 9b shows a schematic plan of the architectural structure of the *huaca*. About the lifecycle (Figure 9a), we focus on the addition of the modeling of conservative cover as a type of conservation intervention to protect bas-relief. An arrow indicates the semantic relationship that links the two entities in the 3D BIM model.

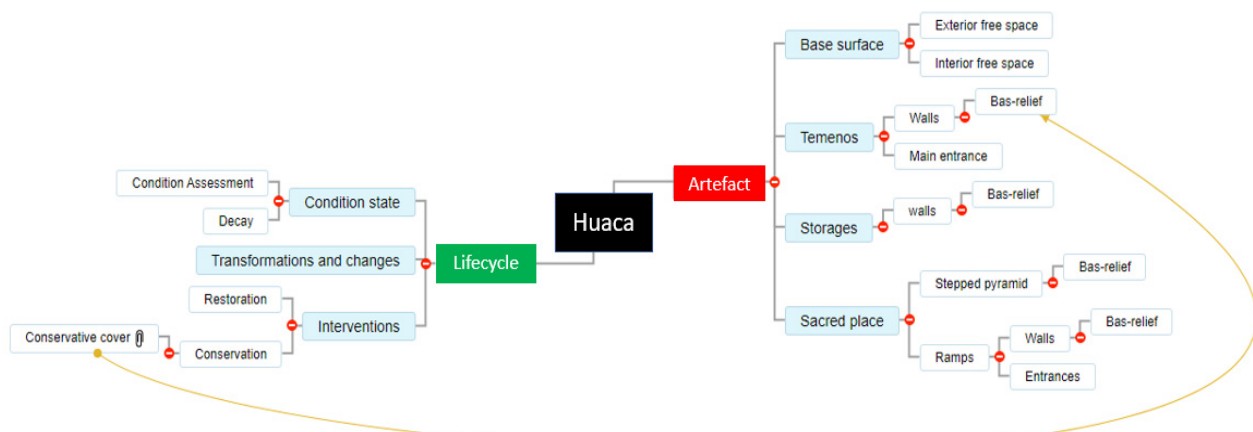

**(a)**

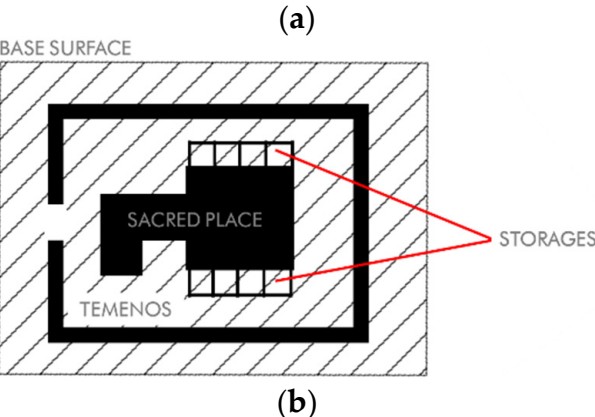

**(b)**

**Figure 9.** (**a**) Semantic ontology of artefact domain referring to the *huaca*. (**b**) Semantic architectural breakdown of the huaca referring to the artefact ontology.

Starting from this schema, other aspects of the ontology can be analyzed and implemented, such as elements from the investigation process and the current state of preservation of the huaca, e.g., the type or level of decay.

At the end of the ontology-based modeling, the problem of merging this conceptual scheme with the HBIM working space arose. As interoperability between ontology and BIM software is not yet an automatic process, the ontology was implemented directly in Autodesk Revit (used as BIM software), leveraging the general three-level hierarchy of both modeling systems: *class > sub-class > entity* for the ontology environment and *family > family type > instance* for the BIM one, as is described in [44] (Figure 10). In this way, the ontology-based modeling was not imported into Revit but served as a guiding principle for the 3D modeling and the subsequent phase of data enrichment.

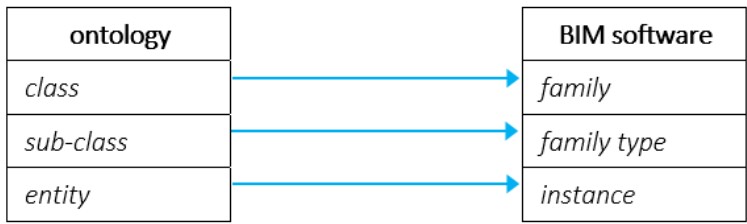

**Figure 10.** Three-level hierarchy of ontology according to BIM information structure.

## 3. Results

### 3.1. HBIM

This section describes the HBIM applied to the built heritage represented by the Huaca Arco Iris. Starting from the data acquisition with geomatic techniques and data search related to historical and technical analysis, and from the definition of the main ontological domains, all these data can be managed within the information system.

For the HBIM approach two steps are followed:

- first of all, a more specific ontology is built to define the architectural components to be semantically modelled and the attributes are linked to them based on the hierarchical structure typical of Revit software;
- then the ontology-based modeling phase of the geometrical 3D model of the *huaca* is developed. First, the parametrization of the 3D shapes is generated within Rhinoceros software, then the 3D objects are imported to Revit together with the definition of the related object family.

#### 3.1.1. Information Management System

With the structuring of the basic ontological model of the *huaca*, we then proceeded to define the structural components and the geometric and non-geometric attributes related to them. Based on the tripartite structure typical of Revit software, a relational database was created to highlight the syntactic structure and semantic relationships within the various families.

Microsoft Access was used to create the relational database to manage all the information referring to each spatial and constructive component of the huaca. The relational database can be accessed through the Autodesk BIM 360 of ULima platform, provided as online Supplementary Material to the paper. As explained in Section 2.3 above, the ontological schema shown in Figure 11, referring only to the *temenos* element, was developed based on the typical hierarchical structure of a BIM software. Each title of the box represents a *family* component, and the list below consists of *type* and *instance* information. Semantic relations among the elements are shown. These links are based on the same identification code (ID) assigned to the architectural component.

#### 3.1.2. Semantic 3D Modeling

After the ontology structure was defined, the next step consisted of creating the semantic ontology-based HBIM model of the *huaca*.

The process of parameterization of geometric shapes such as the meshes of the bas-reliefs was first performed with Rhinoceros software (version Rhino 7), which allowed for correction and simplification of the point cloud coming from the photogrammetric restitution (Figure 12). According to the ontology schema of the artefact (Figure 9a), the 3D shapes of the architectural components are modelled.

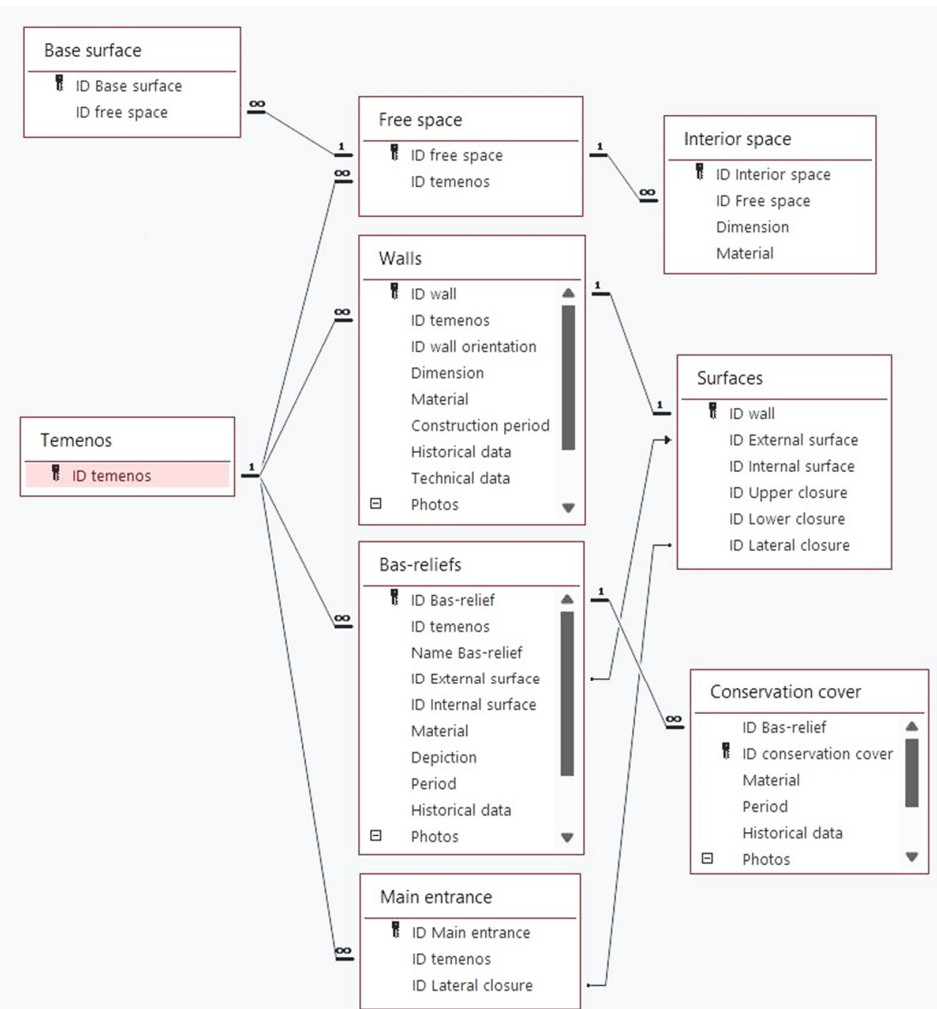

**Figure 11.** Structure of ontology in referring to the element *temenos*.

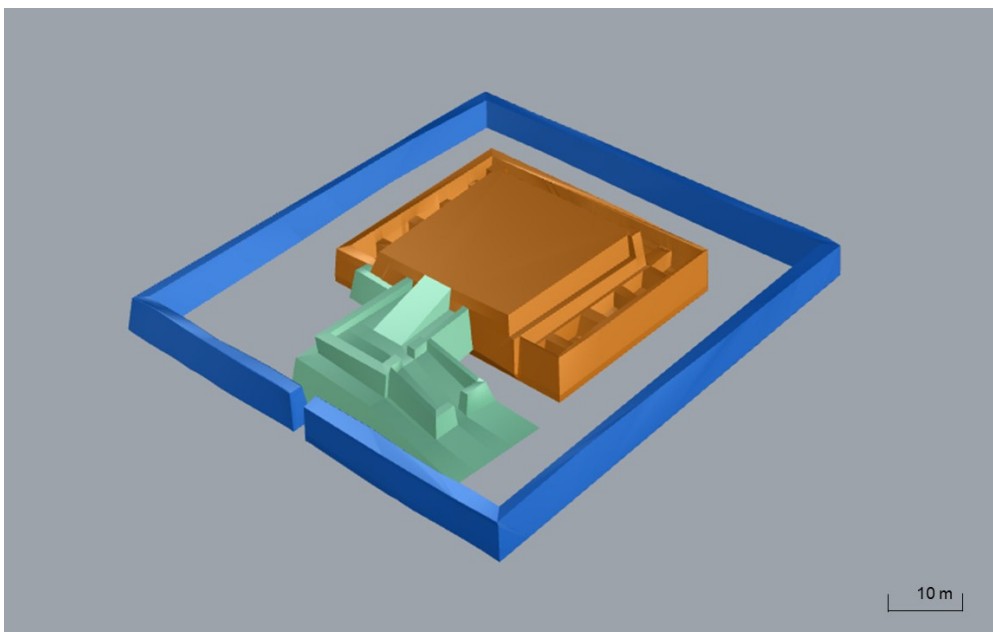

**Figure 12.** Ontology-based 3D model of *huaca* in Rhinoceros.

Meshes of the bas-reliefs and conservative cover are managed to obtain the correct alignment with the relative walls (Figures 13–15).

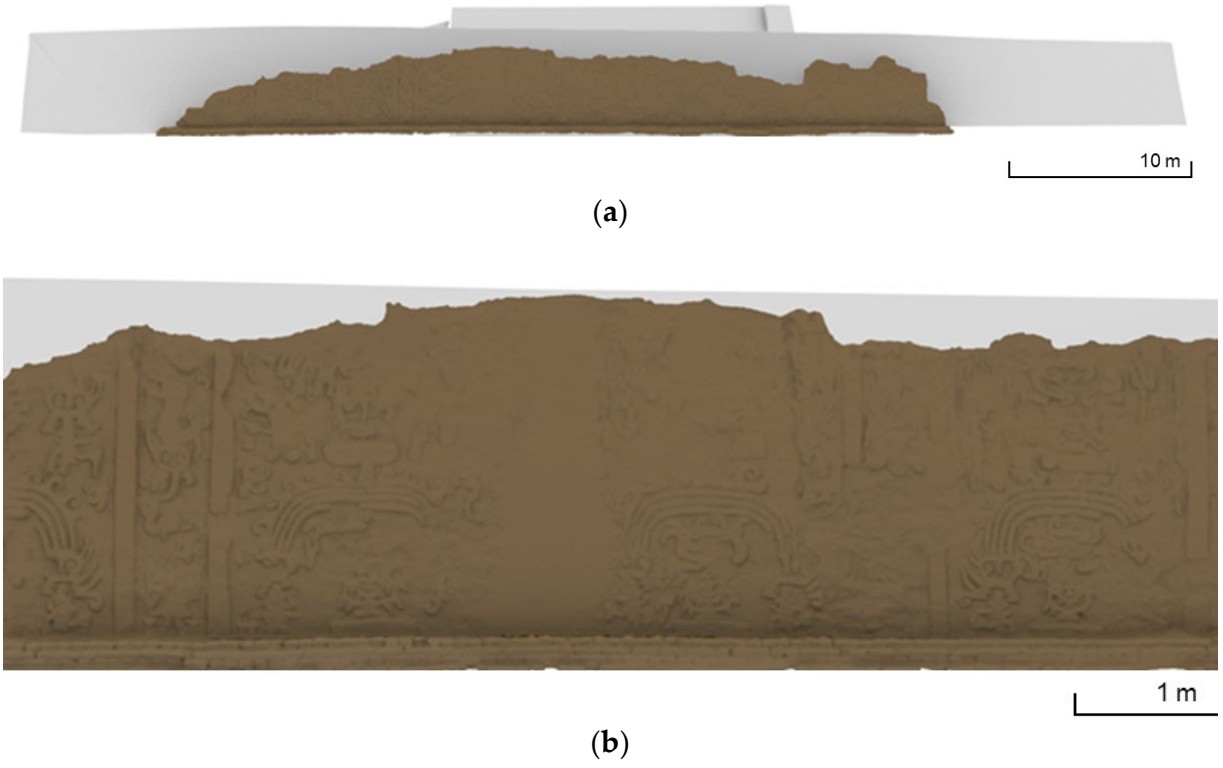

**Figure 13.** Textured mesh of bas-relief (**a**) placed on the wall; (**b**) detail.

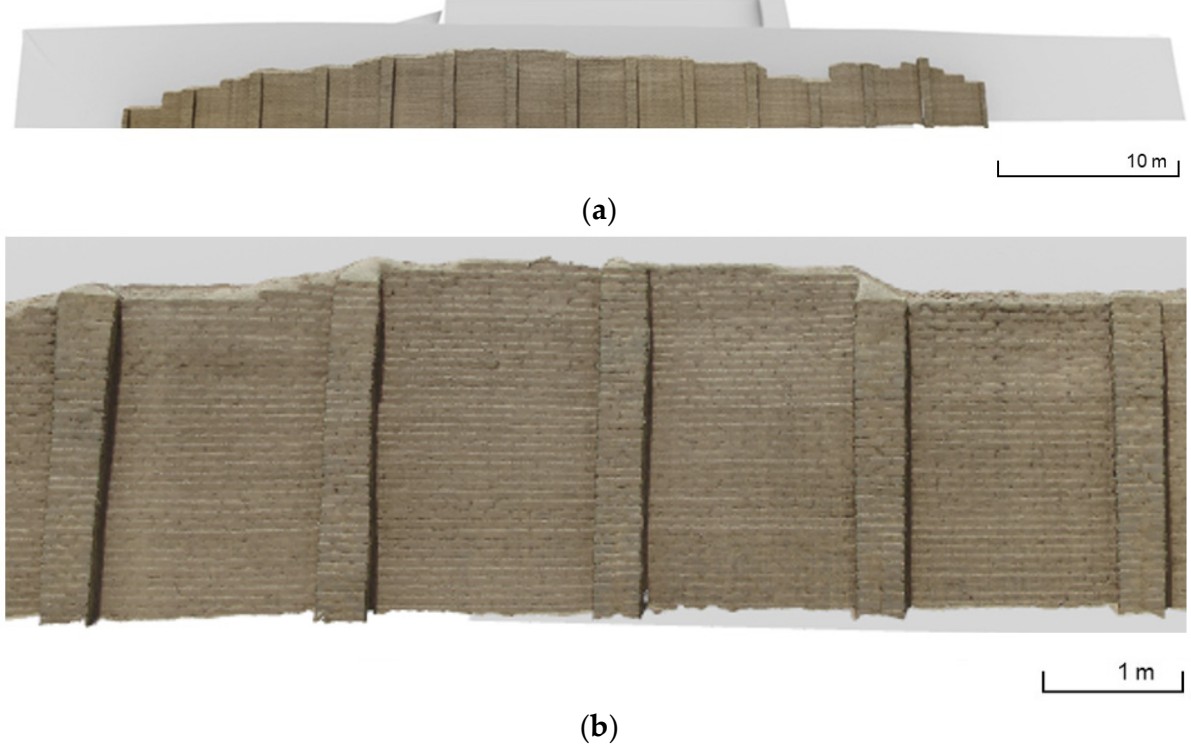

**Figure 14.** Textured mesh of protective cover superimposed on the above bas-relief (**a**) placed on the wall; (**b**) detail.

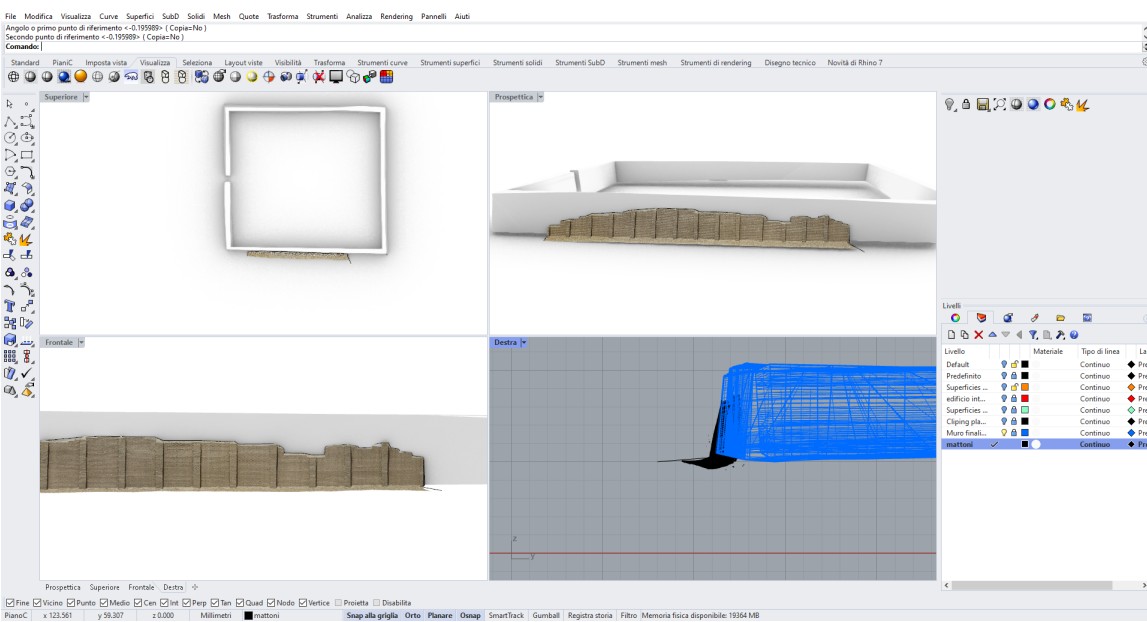

**Figure 15.** Modeling operation of the protective cover, in Rhinoceros.

At the stage of importing the geometric objects into Revit software (version Autodesk Revit 2022), we took into account the ontological schema for each individual architectural component of the huaca in order to then be able to define the various families of elements in the 3D model.

According to the semantic structure of the architectural components belonging to the element *temenos*, the relationship wall/bas-relief/protection cover is tested within Revit.

First the general family 'wall' is created, followed by the definition of the external surface as a sub-family (Figure 16).

Since the 'wall' element is already provided by the BIM software, as for the 'bas-relief' and 'protection cover' elements represented as new families, they must be imported as an external file in the form of loadable families. This is why we speak of HBIM as a customized approach for BIM applied to the built heritage when there are no families of architectural elements characterizing the style of these historical artefacts.

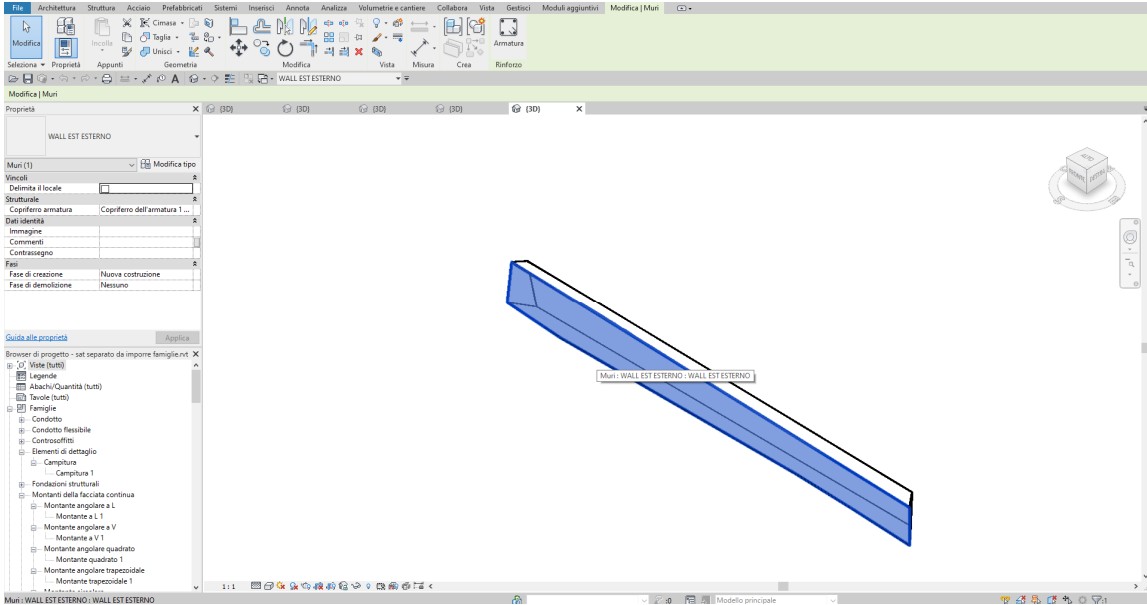

**Figure 16.** Definition of the sub-family "external surface" of the wall, in Revit, which will contain the related bas-relief and protective cover.

An axonometric view of the three elements related to each other is shown in Figure 17, but in graphically defining this relationship we make sure that the 3D model of the bas-relief and the protective cover intersect the surface on which they are placed. This creates *nested families*. The nested family is a particular type of family that contains other separately listed families. This is useful both for creating general parameters applied to all individual elements and for querying information, since it behaves as a whole, within abacuses. If one formulates a query to the external surface of the wall it will show that, in addition to belonging to the temenos wall, it also contains the families of the bas-relief and the protective cover.

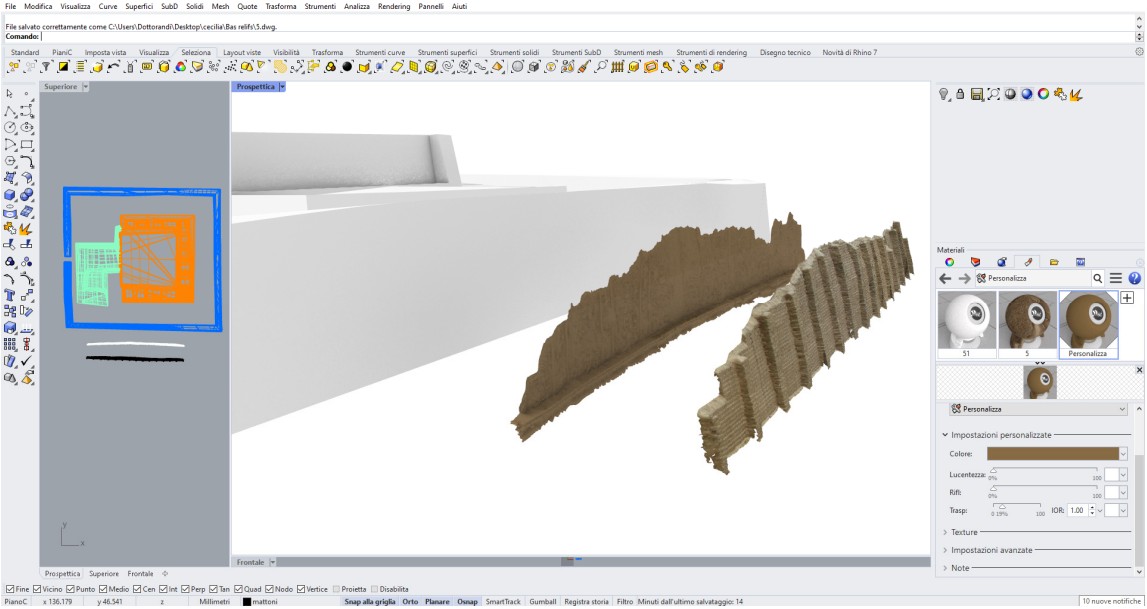

**Figure 17.** Axonometric view of the relationship between external surface of the wall of the temenos, the bas-relief and the conservative cover according to the ontology illustrated in Figure 10.

The ontological aspect of the 3D semantic model consists of the definition of relationships between spatial entities. Formulating a query in BIM, the output consists of a table displaying which family the entity belongs to and what its connections are. For example, if one formulates a query about the conservative cover, the answer is its location. In the case shown in Figure 17, it is located above the bas-relief of that specific wall surface belonging to the temenos. Figure 18 shows the ontological relationship, based on the entity ID code annotation, referring to Figure 17.

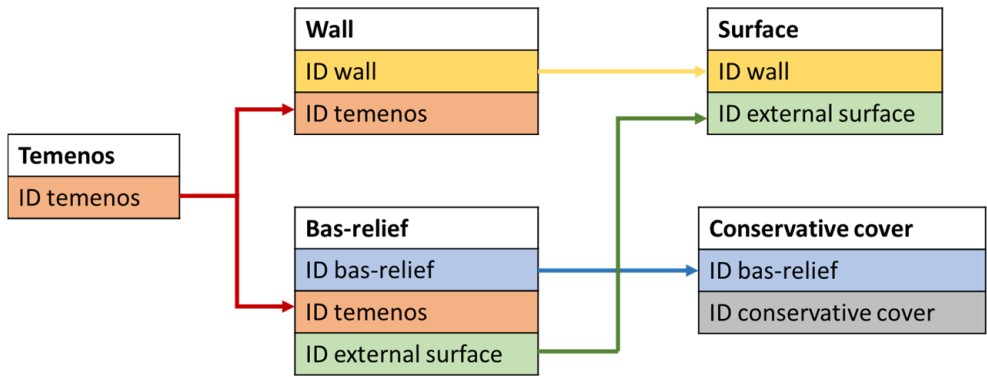

**Figure 18.** Part of the ontological schema, according to Figure 10, of the relationship, based on ID annotation, between external surface of the wall of the temenos, the bas-relief and the conservative cover.

## 4. Discussion of Results

The creation of an HBIM to the archaeological heritage built in raw earth was prompted on one hand by the specific construction technology and its variants and, on the other, by the considerable and almost infinite expressive variability of its artefacts. Parametric modeling, which is the basic procedure for the construction of an HBIM, represents a real challenge in the case of heterogeneous and irregular structures such as those of raw earth, where the manual realization of the elements, and therefore the extreme variety of measures and shapes, prevails.

In the case study presented here, leaving aside the construction technique of compacted clay and mud walls with support of wooden braid or reeds, we focused on the construction technique of hand-made and sun-dried brick (adobe) with rare inserts of wood or stone pebbles. The most complex aspect to deal with was the great figurative variability of the earth architecture that rarely provides sharp edges at the meeting of masonry, but more often rounded corners whose shapes and dimensions do not follow precise geometrical rules. The complex variability of architectural forms is drastically reduced for Chan Chan and for many complexes built in earth, in which the arch and vaulted roofs are virtually absent. Considering the abovementioned challenges, digital photogrammetry proves to be the optimal technology for documentation of this kind of archaeology, since it warrants a good balance between costs, accuracy and detail of the information collected.

Nonetheless, the management of heterogeneous information requires not only a high geometrical detail, but even the definition of a data structure that still does not have its own standard. This is essential, since HBIM platforms have become an indispensable tool for the analysis and management of monuments because they transform the 3D model into a real three-dimensional database. Historical and architectural data, together with data on materials and their composition, conservation status and management measures, coexist.

This work moves in this direction, by defining an ontology for architectural built heritage. Once we defined the architectural and constructive elements, we began the ontological representation of its three-dimensional model, limiting the process to the constructive and geometric aspects defining the intersection of the walls with artificial regular lines instead of the original rounded shapes.

In fact, the realization of an HBIM on archaeological monument requires the necessary simplification of the three-dimensional model and its geometrical forms.

In our case, as in others in the literature, imperfections, deformations or curvatures not perceptible to the naked eye, have been neglected [45]. A representation with a high level of geometric detail would have involved the generation of overly heavy files, thus compromising the possibility of using BIM for monument management activities. Therefore, considering our ultimate purposes for the realization of the model, it was expressly chosen to reach an average level of geometric detail, deepening instead, the enrichment of the information level.

To overcome the state of art limitation of HBIM in archaeology, our contribution lies in the ontology-based information system, which creates a relation between objects, information and geometries; compared to the literature, the most common approach is entrusted to a simple data layer [46,47] neglecting the semantic relation among elements.

Another important aspect to be considered is that the plurality of the working group is given by the collaboration of several public and private institutions. This includes the main Italian public research institute (CNR), an Italian Polytechnic University, two Peruvian universities and the Peruvian Ministry of Culture.

This heterogeneity responds to the first Principle of the Seville Charter, which recommends the multidisciplinary nature of the intervention on cultural heritage [48].

The BIM environment being a bottleneck for the real exploitation of data (only domain experts have the skills to work in such environment), our solution overcomes this limitation, adopting a shared web-based platform where all the involved actors and stakeholders

can contribute with their knowledge. Off the shelf BIM environments are not designed for cultural heritage, and the added value of our ontology-based solution lies in the interoperability and scalability for other case studies, in addition to the expansion of the Huaca Arco Iris. HBIM offers the possibility of exploiting this platform for the management of its vulnerabilities by also involving other actors, such as archaeologists, in the analysis of the artifact in its completeness and complexity.

Although this work shed light on the importance of adding semantics to the HBIM, it is worth pointing out a few limitations. First, the parameterization of very irregular and poorly defined forms required a lengthy work of methodological refinement; the process is neither automatic nor straightforward, and that may be useful for dealing with similar cases spread around the world. This experiment is limited to a small number of elements and information, but we are confident that in the near future we can easily include other features for the monitoring of structures and bas-reliefs, conservation and enhancement actions, identification and prediction of catastrophic events, including simulations and experimental tests [49]. The ontology, even if proved and verified on a real test, is in its infancy and requires more work, especially as concerns non geometric attributes in association to the building elements modelled following the ontology schema itself. Additionally, for the specific case of mud archaeology, the definition of a degradation ontology is paramount, and more work is needed to define the relation between conservation state, change detection over time and the relation with protection structures.

Lastly, we propose to integrate the HBIM model with the 3D GIS of the buffer zone of the Chan Chan Archaeological Complex. In this way we hope to achieve a constant monitoring not only of the monument but also of the territory in which it is inserted, producing an environmental and anthropic risk map, which represents one of the main objectives of our bilateral project.

## 5. Conclusions

An ontology-based HBIM for mud archaeology has been presented. The Chan Chan Archaeological Complex, unique in the world for its size, is subject to several factors of degradation. The monuments are affected by severe material degradation: sea salts rising by capillary action; oceanic wind; and el Niño weather phenomenon. Chan Chan is also vulnerable to environmental and anthropogenic degradation. Its proximity to the constantly expanding city of Trujillo represents a real threat to the archaeological site and its territory.

Our bilateral project aims to contribute to the preservation of the monumental complex through the identification of the main risks and vulnerabilities and through the activation of a constant monitoring of the monuments and the territory.

We believe that HBIM systems can be very useful for achieving these objectives, especially if they are related to GIS and 3D GIS platforms for spatial data management [50]. All monuments of Chan Chan are built in adobe, as well as many other monumental complexes of the American continent, so our indications, once perfected, may constitute a valuable monitoring tool for much of its historical heritage and, in particular, for the management of large monumental complexes.

Our work therefore takes a first step towards the methodological development of BIM on monuments in earth with the belief that it can be of support not only to the administrators of the territory of Chan Chan, but more generally to researchers and institutions dealing with the vulnerability of mud archaeological heritage in various parts of the world.

**Supplementary Materials:** The following supporting information can be downloaded at: https://hdl.handle.net/20.500.12724/16358 (accessed on 7 June 2022).

**Author Contributions:** Conceptualization: F.C., F.J.L.T., E.S.M. and R.O.; methodology: F.J.L.T., E.S.M., F.D.S. and R.P.; software: F.D.S.; validation: F.C., F.J.L.T., E.S.M. and R.O.; formal analysis; F.C., F.D.S., F.J.L.T., E.S.M., R.O. and R.P.; investigation: F.C., E.S.M., R.O. and R.P.; data curation: F.D.S.; writing—original draft preparation: F.C. and F.D.S.; writing—review and editing F.C., F.D.S.

and R.O.; project administration: F.C. and F.J.L.T.; funding acquisition: F.C. and F.J.L.T. All authors have read and agreed to the published version of the manuscript.

**Funding:** This research was funded by Consiglio Nazionale delle Ricerche (CNR, Italy) and Consejo Nacional de Ciencia, Tecnología e Innovación Tecnológica (CONCYTEC, Perú) in the frame of the bilateral agreement for scientific collaboration 2021–2022. The project received the contribution of the Italian Ministry for Foreign Affairs and International Cooperation and of the Instituto de Investigación Científica de la Universidad de Lima.

**Institutional Review Board Statement:** Not applicable.

**Informed Consent Statement:** Not applicable.

**Data Availability Statement:** The data are available upon request on the platform Autodesk BIM 360 of ULima. You can contact the corresponding author or Francisco James Leon Trujillo.

**Acknowledgments:** We would like to thank the Ministry of Culture of Peru and the Proyecto Especial Complejo Arqueologico Chan Chan (PECACH), who have always supported our searches on the site. We are very grateful to the Italian Ministry for Foreign Affairs and International Cooperation for its institutional recognition and financial contribution. We would like to express our gratitude to the Istituto Italiano di Cultura of Lima for supporting us during our mission in Peru. Moreover, our thanks must go to Carlos Antonio Espinoza Brugman and Frank Kevin Neri Caipo, graduate students of Universidad de Lima, for their elaboration of the model in Rhinoceros, and to Cecilia Ruggieri, student of Università Politecnica delle Marche, for her help in importing the geometric objects into Revit software.

**Conflicts of Interest:** The authors declare no conflict of interest.

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
