# Peer review of "Exploiting HBIM for Historical Mud Architecture: The Huaca Arco Iris in Chan Chan (Peru)"

_heritage, doi:10.3390/heritage5030108_

Round 1

Reviewer 1 Report

The manuscript is an important contribution to the studies of the BIM usage in archaeology, it significantly broadens the study described earlier in the paper cited herein as [34], adding substantial amount of knowledge related to the use of ontology in HBIM applications. In my opinion, the manuscript should be therefore published. Minor changes are required to ensure that the data is available, the sites mentioned in the text are illustrated on the map, and editing issues and/or typos (very few) are removed. I have read the paper with great interest.

The merits

line 195 - You refer to your earlier published contribution ([34]), but consider adding here a sentence or two regarding the state of the research before you started your present work, and point out new additions that justify the importance of new paper on Arca Iris. In my opinion, it is obvious that the manuscript significantly broadens the scope of earlier research, which was confined to 3D reconstruction of the huaca. But this can be unclear to reader, who can ask himself if he finds any new content in the paper.

Figures 8-10 - Here you explain the structure of the ontology used to create a HBIM model of the huaca. In my opinion, this is the most important outcome of the paper. The ontology is well-documented, but for the reader it can be difficult to grasp the entire ontology schema from the separate figures. Please consider adding a supplementary material which would include an example of the record of the relational database with data regarding components of the huaca. I know that reconstructions are available for the reader through an online viewer (lines 582-583), but it is the ontology (the backend) which is the most important here. An online supplementary material would significantly improve the data availability and scientific soundness of the manuscript.

Minor issues

line 71 - I know that BIM/HBIM acronyms are widely recognised and accepted, but I would recommend to add an explanation in brackets where this acronym appears for the first time, either here or in the abstract.

Figure 1 - The map is important, but it could provide more information. Please think about replacing it with small map of the entire Peru, with an inlay showing the vicinity of Trujillo (mentioned in line 150, but not shown on the map) and Chan Chan, and other locations mentioned in the text (see my next comments). It is also important to add the description of the photograph in the figure caption. Please remember about adding longitude, latitude and scale of the map.

line 185 - Arco Iris is not shown on the map. Please include it in Fig. 1.

lines 221-224 - Huacas other than Arco Iris are listed here, but they are not included on the map. Figure 1 is the best place to show their locations.

lines 235-240 - You are giving here many details regarding the Arco Iris huaca. Make sure that it is not necessary to refer here to earlier papers which describe the huaca, if there are any.

lines 374-387 - You are starting the Results section, but for me it looks like description on the method you use (the steps that are required to employ HBIM). Consider moving this subchapter to the Methods.

lines 469-473 - Are you sure that the results and discussions are clearly separated? This paragraph looks like integral part of the discussion, not the Results section.

Editing issues

line 5 - missing comma after "Roberto Pierdicca"

line 68 - use "technical" instead of "technicalto"

line 85 - I am not a native, but I've noticed that names of scientific disciplines are not capitalised; use "archaeology" instead of "Archaeology"

lines 282-283 - I don't think that new paragraph is necessary here. Although the the text does not require substantial proofreading, I would recommend removing short paragraphs which include one or two sentences directly linked to earlier discourse. For example, see lines 481-495: you will find there three paragraphs, each of them includes a single sentence.

line 518 - The term "huaca" is capitalised here, but in its earlier appearances you used lower case letters. Please be consistent.

lines 553-559 - The word "subject" is repeated here three times. I know that repetitions are necessary to ensure the text is strict and scientific, especially in the conclusions, but please make sure that it is not possible to use synonyms here.

line 647 - insert missing space ("BIM-enabled heritage")

Reviewer 2 Report

This paper refers to the use of HBIM for buildings in raw mud. The authors use the data coming from the direct and indirect analysis, and an ontology is built to guide the management of these data within an HBIM system.

The subject is interesting and the structure of the manuscript is well defined. Please add a paragraph in the discussion section about the usage of the results (to emphasize the importance of your method).

The conclusion section still does not present a drawn conclusion. Please try to draw a pure conclusion from your own analysis/data/study, which finally could contribute to the Body of Knowledge.

The reviewer recommends that the paper be edited for language. Be consistent on American vs British English, eg. lines 325, 327, 328, 361, 367, 368, 405 and 410: modeling vs line 96, 130, 131, 139, 211, 294, 323, 365, 384, 436 and 479: modelling. Correct miswritten words such as line 27: conteinedàcontained, line 82: aquisitionàacquisition, etc.

line 19: Use the abbreviation (HBIM) only after the full meaning. Change the information system to Heritage Building Information Modeling

Reviewer 3 Report

The authors are congratulated for the document both in its structure and in the fulfillment of the objectives, references, and practical approach to the safeguarding of built heritage in raw earth; however, the only pertinent observations that I will be able to comment on are:

Abstract:

In general, it should be completely restructured and highlighting in a clearer way the objective of the research, the innovation of the method and the advantages over the results and conclusions.

Line 22 - Use of acronyms, not describing what the acronym CNR is.

Line 25 - The support of philosophical elements is interesting; however, avoid falling into rhetorical, conceptual phrases and not being concrete and determinant in the results of the research.

Introduction:

Improve the wording and structure of the paragraphs as it generates very large paragraphs that dilates the objective idea of the reading.

Another recommendation that will allow a logical reading of the document is to place 1.3, which describes the settlement of Chan Chan, before 1.2, which describes the BIM, for order and context.

Line 39 – 41: The paragraph needs to be revised, it has erroneous data for the present, rely on more recent research, sources 3, 4, 5 have as (20, 40 and 27 years respectively) without underestimating that it can be considered gray literature, it is recommended to restructure it.

Line 52: Review and update the recorded data.

Line 71 – 78: Restructure the paragraph, it is confusing, and the ideas are repeated.

Line 72: Innovative for Latin America, however, in Europe this method has been used for several years and is multiplatform, not only linked to AutoDesk as in the present research.

Line 82 – 109: Restructure the paragraph, it is a paragraph too long that confuses the reading, the author should coordinate ideas in shorter paragraphs separating them by applicability, method, examples, and other variables exposed.

Review examples of the research groups of the University of Seville, Polytechnic of Valencia, Polytechnic of Bari, Polytechnic of Milan, and their progress in graphic expression applied to BIM, HBIM and AHBIM.

Line 110 – 125: Restructure the paragraph, it is a long paragraph that dilates the idea of the research.

Line 130 – 146: Restructure the paragraph, it is a long paragraph that dilates the idea of the research, it does not comment on AHBIM as a method for compiling information, very applicable for the purpose of the research.

Figure 1: This figure would be of more help for the location if, in addition to observing a modest general location of the context, a composition of what is described in the previous paragraph could be generated.

Line 161 – 173: Revise the paragraph has drafting errors.

Line 188 – 196: Revise the paragraph, because it repeats information from 1.2 in the usefulness of the methodology, considering that there are more contributions and uses of the HBIM methodology that address in the information and its contents.

Materials and Methods

Common problems in paragraphs

Line 198 – 211: Restructure the paragraph, it does not communicate anything to me; be cautious with the use of the term ONCOLOGICAL because the methodological process of BIM is already structured, and the technicality confuses the common reader.

Line 217 – 291: It is suggested to the research group that it could unify the information and consolidate it in a single group "introduction" to the points 1.3 Research purposes and aims 2.1 Case study: huaca Arco Iris in Chan Chan. They describe the same.

Line 295: This method of data extraction "spherical photogrammetry" subtracts a lot of information to the current research where the quality of the data allows the three-dimensionality and information of these.

Line 311 – 314: The method needs more information such as the protocols followed, number of samples, quality - resolution, technical equipment for sampling, simple and mosaic image processing software, coefficients and variables of loss or diffusion of the graphic acquisition, etc.

Line 319: Review if relevant this subsection "2.3 Dedicated semantic ontology".

Figure 7: Would be more interested in describing the methodological process associated with BIM than what the graphic describes.

Line 361 – 368: Restructure the paragraph

Figure 8: Revise the image described.

Figure 9: Improve and revise the image described.

Results

This section should be rewritten more clearly, it is recommended to restructure it, there are ideas that are repeated in other paragraphs, tell us the result of the applicability of BIM as a benefit for the built heritage in raw earth, the contribution of Lines 381 - 387 is less than modest for the information and data benefits generated by BIM and the platforms associated with its use.

Line 405 – 415: Revise the paragraph, Restructure the paragraph

Figure 11 - 12 - 13 - 14: Use the benefits of the software to generate a picture that impacts the use and applicability of the BIM methodology.

Line 438 – 459: Restructure the paragraphs on these lines.

Discussion

Revise the structure and wording of Line 476 - 550 (Restructure the paragraph)

Conclusion

Do not generate a contribution.

References

Review the references used, although there is a considerable number of sources from the last five years, the use of sources of very little justification in the scientific work and the benefits of BIM for patrimonial purposes is very detrimental to the objective of the research.

Reviewer 4 Report

Review of heritage 1785844

 Exploiting HBIM for historical mud architecture: the huaca Arco Iris in Chan Chan (Peru)

The paper needs to be much  better referenced. There many assertions made THROUGHOUT that are not backed up with references as would be required  for an academic paper.

It is an interesting paper that with some alterations is worth publishing. One of the key issues I would like to see addressed is how the various ontological aspects can be used in the model and visualised. 

Also showing original vs intervention would be useful.

Line 228          why is it an intangible area?

Line 270          what are ‘real remakes’ ?

Line 335          No, the artefact itself has ontic properties (geometry, materials etc) while the rest are ontological properties

Figure 7, actors: what about the people built the site? What about people that used the site? While they are no longer , they too were actors. Mention and exclude them

Figure 7 stakeholders? Who are they? Are experts not stakeholders too? This needs to be MUCH better though through

Figure 7 in the sphere of investigation process the evaluation of the site in terms of meaning to the current residents of Trujillio is lacking, as is the evaluation terms of meaning to tourists (who are also actors)

Did they include historic imagery of past states in the system and if where and how. Why only in some and not in all?

MINOR ISSUES

Figure 1           A full width image of the inset would be useful as a new figure

Figure 3 needs much better explanation and sub-numbers

Figure 7, there is  a stray box with “…’ under actors

The paper needs a through edit by a PROFESSIONAL native-English speaking EDITOR. There are numerous grammatical issues as well as inappropriate choices of words (eg. line 40 ‘goods’ instead of ‘places’ or “Since the 90s of the last century..”

Round 2

Reviewer 3 Report

The authors should be congratulated for improving the document both in its structure and in the recommendations mentioned above, however, the only outstanding observations that I can comment on are:

1.- The introduction is clearer and describes the place of study well.

2.- Better describe the methodology and clarify how the nature of being "ontology" is related to the organization of BIM methodology, which is a structured process and why this research emphasizes it.

3.- The graph of the images such as Nº5, Nº6, Nº7, Nº12, Nº13, Nº14, Nº15 must be supported by an indicator or graphic scale; while figures such as No. 8, No. 9, No. 10, and No. 11 should indicate having a clearer composition support than a “Screen Print”, improving the graphic composition of images, tables, and figures.

4.- The bibliographical references have been increased and revised, realizing that No. 4 does not have a citation.

Reviewer 4 Report

The authors have undertaken some revision , but this does not go far enough

In my initial review I wrote: It is an interesting paper that with some alterations is worth publishing. One of the key issues I would like to see addressed is how the various ontological aspects can be used in the model and visualised.

The authors responded:

we have tested the only ontological relationships between the construction entities in the 3D model of huaca, eg. external wall -  bas-relief – conservative cover. 

Further comments: 

The authors have not addressed the issue: how the various ontological aspects can be used in the model and visualized. Any object has two dimensions Ontic (i.e. dimensions, colour, material) and ontological. Ontic elements are easy to database and visualize. But while the ontological elements are commented on throughout the paper but are only cursorily addressed in the paper it is unclear how the ontological elements contributed by actors, for example, can be and are visualized. This needs to be discussed and exemploified

=--------=--------=--------=--------=--------=--------=--------

In my initial review I wrote:

Also showing original vs intervention would be useful.

The authors responded:

It is an on-going work, ….Future work consists of the creation of shared parameters regarding non geometrical information related the each 3D object of the huaca and also the representation of degradation forms.

Further comments: 

This does not explain why the original vs intervention stage are not visualized.

=--------=--------=--------=--------=--------=--------=--------

In my initial review I wrote:

§  Figure 7, actors: what about the people built the site? What about people that used the site? While they are no longer , they too were actors. Mention and exclude them

§  Figure 7 stakeholders? Who are they? Are experts not stakeholders too? This needs to be MUCH better though through

§  Figure 7 in the sphere of investigation process the evaluation of the site in terms of meaning to the current residents of Trujillio is lacking, as is the evaluation terms of meaning to tourists (who are also actors). Did they include historic imagery of past states in the system and if where and how. Why only in some and not in all?

The authors responded:

Figure 7 show a general ontology referred to built heritage. It is not complete, can be filled with other terms, such as other actors involved. But the actors showed in the graph are only those are involved within the work related to the BIM application of the huaca. As reported in the reference [34] the involvement of other people, like local administration or residents of Trujillo, was useful to test the exhibition created for the museum of the huaca, also with a virtual tour experience. The objective of this work is only the definition of BIM platform to manage all the information managed by experts in the fields and then, as future work, can be shared to other actors can be involved.

Further comments: 

If the authors do not want to address this in detail (which I recommend to do), then they need to  make it clear to the reader that they are aware of these limitations and state them as such.
